

# In situ measurement of meltwater percolation flux in seasonal alpine snowpack using self potential and capillary pressure sensors

Wilson S. Clayton

Department of Civil and Environmental Engineering, Colorado School of Mines, Golden, Colorado, 80401, USA

*Correspondence to*: Wilson S. Clayton (wclayton@mines.edu)

**Abstract.** Downward flux of percolating meltwater was measured quantitatively in an in situ vertical profile, in an alpine snowpack, at a remote location. Three separate measurement systems were used to obtain multiple parameters required to calculate percolation flux. Brooks-Corey constitutive parameters were measured in a 0°C isothermal snow sample test cell, and then applied to an on-site snow column test. The instrumented column test allowed calculation of fluxes, that were then

calibrated to measured outflow to empirically determine an appropriate value of zeta potential. In situ measurements with data logging of self-potential (SP) and capillary pressure sensors then allowed calculation of flux from SP measurements ($q_{sp}$), expressed as darcy velocity, over a multi-day period. The results strongly reflected diurnal snow melt dynamics, and daily peak $q_{sp}$ ranged from 5.6 to 105 cm/d. $q_{sp}$ was comparable to actual fluxes, represented by changes in snow water equivalent (SWE) (2.5 to 5.3 cm/d) measured at an adjacent USDA SNOTEL station. The average error in $q_{sp}$ was 8% over a four-day

period, with total calculated flux of 18.1 cm, compared to a 16.8 cm change in SNOTEL SWE. Daily (24-hour period) errors ranged from +26% to -47%. The methodology developed herein can combine SP with either capillary pressure or saturation measurements. The ability to measure meltwater percolation flux in snowpacks may support mathematical modeling of unsaturated flow processes in melting snow, and may supplement studies of snowmelt-groundwater and snowmelt-runoff interactions and glacier mass balance studies.

**1 Introduction**

Approximately one sixth of the world's population relies on snowmelt or glacier runoff for their water supply, and in these areas the timing and magnitude of snowmelt exerts a major influence on water availability, drought, and other potential effects of climate change (Barnett et al., 2005). For seasonal snowpacks, the timing, rate, and magnitude of meltwater percolation and retention in the snowpack are important factors controlling recharge of groundwater systems and surface runoff

(Musselman, et al., 2017, Smith et al., 2014). For glaciers, percolation of meltwater is an important factor affecting glacier mass balance (Jansson et al., 2002), particularly with respect to the lag between melting and percolation, the retention and storage of meltwater within the firn, and the delayed release of meltwater deeper into the glacier hydrologic system.



Colbeck (1972) first applied then-existing multiphase flow theory to the physics of unsaturated flow during snowmelt percolation and the downward propagation of wetting fronts during diurnal snowmelt cycles. Later, Colbeck (1975) extended the theory to layered snowpacks, and subsequent work (Marsh and Woo, 1985) showed that snowpack heterogeneity leads to flow channeling and large spatial variability in meltwater flux. Mathematical models of unsaturated flow during snowmelt

percolation have increased in complexity over time, incorporating and coupling processes such as thermal effects and re-freezing, capillary forces, hydraulic processes, and multi-dimensional preferential flow (Illangasekare et al., 1990, Tseng et al., 1994, Hirashima et al., 2014, Wever et al., 2014).

The work reported herein was focused on field measurement, and the unsaturated flow parameters of water saturation, capillary

pressure, and flux are desirable to measure for snowmelt percolation problems. Measurement of capillary pressure using porous cup tensiometers was described by Colbeck (1976) and Wankiewics and De Vries (1978), although more recent use is not widely reported. Measurement of water saturation in snow has been accomplished using dielectric methods (Denoth, 1989) and time domain reflectometry (TDR) (Diaz et al., 2017, and Samimi and Marshall, 2017). The use of electrical methods is emerging as a possible approach to measure flux in percolating meltwater. Revil et al. (1999) described a theoretical

framework for the relationship between subsurface water flow and an induced millivolt (mV)-level electrical potential, known alternately as self-potential or streaming potential (SP). Kulessa et al. (2012) extended SP theory to the problem of gravity-driven unsaturated flow in melting snow, and derived Eqn. (1) to describe the SP signal generated by unsaturated flow percolating through a laboratory snow column:

$$\psi_m = \frac{\varepsilon \zeta}{\sigma_w} \frac{S_w}{S_e^n} \frac{1}{kA} Q \tag{1}$$

Where,

$\psi_m$ = streaming potential (V)

$\varepsilon$ = dialectric permittivity (F m$^{-1}$) (constant, at 7.8E-9 Farads per meter)

$\zeta$ = zeta potential (V)

$\sigma_w$ = water electrical conductivity (S m$^{-1}$)

$S_w$ = water saturation

$S_e$ = effective water saturation

$n$ = pore size parameter

$k$ = intrinsic permeability (m$^2$)

$A$ = column cross-sectional area

$Q$ = volumetric flow rate (m$^3$ d$^{-1}$)

Kulessa et al. (2012) verified Eq. (1) by modeling SP signals and calibrating the results to laboratory snowmelt column experiments. Building on the work of Kulessa et al., Thompson, et al. (2016) used SP measurements to assess glacier meltwater





production over a 25 m x 25 m 2-dimensional grid on a glacier surface. They also measured bulk meltwater discharge at a downgradient lysimeter, and found good qualitative temporal correlation between SP signals and diurnal meltwater production. Thompson et al. also evaluated uncertainty and sensitivity in calculation of various parameters in Eq. (1), and concluded that the SP was promising for further, more quantitative development.

For the purposes of the study reported herein, we explore using SP as a field measurement approach to calculate meltwater percolation flux vertically downward through the snowpack under unsaturated flow conditions. Therefore, we can express the darcy flux ($q$, m d$^{-1}$) associated with infiltrating unsaturated meltwater flow, as a function of the streaming potential and a number of other parameters, by rearranging Eq. (1), as follows:

$$q = \frac{\psi_m}{\frac{\varepsilon \zeta}{\sigma_w} \frac{S_w}{k S_e^n}} \tag{2}$$

Where, $q = Q/A$.

The pore size distribution parameter ($n$) in Eq. (1) and Eq. (2) can also be represented in terms of the Brooks and Corey (1964)
pore size distribution parameter ($\lambda$), where $n = (2+3\lambda)/\lambda$. In order to consider the relationships between $S_w$, $H_c$, and relative permeability ($k_r$) the following from Brooks and Corey (1964) are used herein.

$$Se = \frac{(Sw - Sr)}{(1 - Sr)} \tag{3}$$

$$k_r = S_e^{(2+3\lambda)/\lambda} \tag{4}$$

$$k_r = \left(\frac{H_o}{H_c}\right)^{(2+3\lambda)} \tag{5}$$

$$Se = \left(\frac{Ho}{Hc}\right)^{\lambda} \tag{6}$$

where,

$S_r$ = residual water saturation

$H_c$ = capillary pressure head (m)

$H_o$ = air entry capillary head (m)






Equation (2) can be combined with equations (3), (4), (5), and (6), to expressed a flux value calculated from SP measurement ($q_{sp}$), in terms of either saturation (Eq. 7) or capillary pressure (Eq. 8), as follows:

$$q_{sp} = \frac{\psi_m}{\frac{\varepsilon\zeta}{\sigma_w}\frac{S_w}{k(S_e)^{(2+3\lambda)/\lambda}}} \tag{7}$$

$$q_{sp} = \frac{\psi_m}{\frac{\varepsilon\zeta}{\sigma_w}\frac{\left(\frac{H_o}{H_c}\right)^{\lambda}(1-S_r)+S_r}{k\left(\frac{H_o}{H_c}\right)^{(2+3\lambda)}}} \tag{8}$$

Snow density and grain size have a strong effect on $k$, which can be estimated (Kulessa et al., 2012, Colbeck and Anderson, 1982) using the following, from Shimzu (1970):

$$k = 0.077d^2 e^{-0.0078\rho_s} \tag{9}$$

where,

15    $d$ = mean snow grain diameter (m)

$\rho_s$ = snow density (kg m$^{-3}$)

## 2 Objectives

The primary objective of this study was to develop and implement a field-based methodology for the in situ measurement of unsaturated flux during gravity driven percolation in melting snowpacks. The goal was to combine SP measurement with

20    other field measurements, in order to solve either Eq. (7) or Eq. (8) for flux. Use of Eq. (8), based on measurement of capillary pressure, was selected for implementation because $H_c$ tensiometers are less expensive and simpler than TDR technology for measurement of $S_w$. Therefore, this study also aims to assess the preferred method for future work.

A secondary objective of the study was to evaluate unsaturated flow dynamics during diurnal melting cycles in a seasonal

25    alpine snowpack. Temporal changes in flux, capillary pressure, and saturation are expected in conjunction with changes in flow as diurnal wetting fronts move through the snowpack (Colbeck, 1972).





There was an additional objective that all measurements be collected in the field, and that equipment be transportable in a backpack and functional in outdoor environmental conditions. This objective was appropriate to facilitate future snowmelt studies in remote locations accessible primarily by foot.

## 3 Methods

The field study was undertaken at Loveland Basin ski area (Fig. 1) in central Colorado, USA. The snow study plot was located ~1.1 km from the continental divide, at an elevation of 3,475 m above sea level (Fig. 2) and had a slope angle of 8 degrees, a W-NW slope aspect, and was mixed open and forested shade (Fig. 3). The plot was located 30 m. proximal to a United States Department of Agriculture (USDA) Snow Telemetry (SNOTEL) Site. The SNOTEL site provided automated hourly measurement of weather conditions as well as snowpack depth and snow water equivalent (SWE). These data allowed a

comparison of $q_{sp}$ to changes in SWE data during snowmelt. Field studies were completed during May and June 2017.

In order to measure all of the parameters in the right half of Eq. (8), and therefore solve for $q_{sp}$, a combination of three separate but integrated measurement systems were implemented in the field at the snow study plot, including:

15          (a) measurement of capillary pressure - saturation curves in a snow core sample, to determine $\lambda$, $S_r$, and $H_o$;

          (b) conduct of a snowmelt column test that directly measured $q$, $\psi_m$, $H_c$, and $\sigma_w$, which when combined with results from (a), allowed determination of $\zeta$; and

          (c) in situ testing over a multi-day period, directly measuring $\psi_m$ and $H_c$ using in situ sensors, which when combined with results from (a) and (b), allowed determination of $q_{sp}$, within the snowpack.

The methods associated with the three above measurement systems are described in the following subsections.

### 3.1 Capillary pressure - saturation curves

Measurement of capillary pressure - saturation curves was conducted under drainage conditions using a snow core sample collected from a snow study pit, on May 28, 2017. A isothermal test cell (Fig. 4) was used to maintain the snow core sample

at 0ºC in order to prevent melting during the drainage experiment. The snow core sample was 4.2 cm in height and 8.3 cm in diameter, and was collected as an undisturbed snow sample. The measurement of drainage volume (i.e. $S_w$) and applied capillary head ($H_c$) in the test cell were made using a simple graduated manometer system illustrated on Fig. 4. The snow core sample holder included a porous filter at the base with an air entry pressure of approximately 40 cm water to prevent air intrusion into the manometer system. The snow core sample was saturated with 0ºC water, and drained in a number of steps

by applying an incremental increase in capillary pressure and allowing the drainage volume to equilibrate. Capillary pressure - saturation curves obtained were fitted to Eq. (6) to determine the Brooks and Corey (1964) parameters $\lambda$, $S_r$, and $H_o$.





### 3.2 Snowmelt column tests

A snowmelt column test was performed on May 28, 2017, concurrent with the measurements described in Section 3.1. The snowmelt column consisted of schedule 40 PVC pipe with an inside diameter of 0.1 m, and with a beveled end to facilitate coring into the snowpack. The column was driven by hand through the full depth of the snowpack, to the ground surface (Fig.

5). Subsequently, the column was excavated by hand, while completing a snow study pit. The upper third of the column was painted black to provide radiant warming from the sun to drive melting, while mV-level SP signals, and $H_c$ were measured in the lower third of the column. SP electrodes were spaced 0.3 m with the reference (low) voltage electrode on top. The $H_c$ tensiometer was placed at 0.15 m spacing, in between the electrodes. The electrodes and data logger used in the column test were custom fabricated for the in situ measurements, as described in Section 3.3. The $H_c$ tensiometer was obtained from Soil

Measurement Systems, model CL-029A. Column effluent volume was measured over time to determine the actual column meltwater flux ($q_{meas}$) as the column test progressed. Water quality parameters pH and $\sigma_w$ were measured in column effluent water using a Hanna Instruments model 98129 tester, with in-field 2-point calibrations. The column weight after completion of the snowmelt experiment was used to measure snow density.

### 3.3 In situ measurements

SP electrodes and $H_c$ tensiometers were installed horizontally in the sidewall of the snow study pit (Fig. 6) on May 28, 2017, at depths shown on Fig. (7). The SP electrodes were fabricated using 8-mm diameter round Ag-AgCl sintered electrodes (BioMed Electrodes, model BME-8) embedded in epoxy, at a 45-degree angle, in the tip of a 9-mm diameter, 1.2 m long fiberglass wand. The electrodes were inserted into the snowpack to a distance of 1.2 m beyond the snow study pit sidewall. SP electrodes were spaced 0.3 m vertically, and a bubble level was used to place the electrodes horizontally into the snowpack

so that the desired spacing was maintained at the electrode tip. The $H_c$ tensiometer (Soil Measurement Systems, model TM-31) was placed at 0.15 m spacing, in between the lower interval electrodes.

The snow study pit was backfilled after installation of the measurement sensors, and data acquisition was subsequently carried out over a 19-day period during the spring snowmelt cycle. Data acquisition for SP mV-level electrical potential and the $H_c$

tensiometers was accomplished using a Campbell Scientific CR-10 data logger with 12V power supply, housed in a waterproof Pelican Case. In-situ SP and $H_c$ data acquired using the CR-10 data logger were averaged and stored at 10-minute intervals. The value of $q_{sp}$ was calculated for each data logger time step from the SP and $H_c$ data using Eq. (8) and the parameters determined as described in Sections 3.1 and 3.2. The integral of $q_{sp}$ over each 24-hour period ($\int q_{sp-24}$ (m)) was then compared to the 24-hour change in SWE measurements ($\Delta SWE_{24}$ (m)) at the adjacent SNOTEL station in order to assess the accuracy of

the method.





## 4 Results

### 4.1 Snow pit profile results

Fig. 7 shows a log of the snow study pit, prepared on May 28, 2017 at 08:00. The overnight low temperature was -6°C, the ambient air temperature at this time was 4°C, and the study pit was shaded. The total snow depth was 125 cm, over a mixture

of grass and ground cover vegetation. The temperature profile shows an isothermal 0°C snowpack below 104 cm, and frozen snowpack above 104 cm, at the time the pit was logged. All water observed in the snowpack was pendular, no free water could be made to drip from any sample. Snow crystal forms observed below 94 cm were clustered rounded grains with an average diameter of 1.5 mm, with interspersed ice lenses as shown. Two 1 cm thick ice layers were present in the bottom half of the snowpack, although it is not clear if the ice layers were continuous over other areas of the snow study plot including the

point of SP electrode placement, ~3 m distant to the study pit. Depth intervals are indicated on Fig. 7 for installation of SP electrodes and a $H_c$ tensiometer used for measurements reported herein.

### 4.2 Brooks-Corey parameters

Fig. 8 shows the $H_c$-$S_w$ data collected using the isothermal snow core sample test cell, as well as a best fit line from Eq. (6), which provides values for the Brooks and Corey (1964) parameters of $\lambda = 2.7$, $S_r = 0.15$, and $H_o = 0.098$ m.

### 4.3 Snow column test results

Fig. 9 shows transient data collected during the snowmelt column test over a three-hour time period from 14:00 to 17:00 on May 28, 2017. The snowmelt column operated for approximately two hours prior to the data shown, during which time quasi-steady melting and percolation conditions were established. $H_c$ measurements were inaccurate and are not shown for portions of the column test, due to difficulties maintaining intimate contact of the tensiometer porous ceramic cup with the melting

snow in the column. The available $H_c$ data in Fig. 9 show that the capillary pressure was near the air entry pressure during the period of induced snowmelt percolation through the column (i.e., $H_o/H_c \sim 1$), which reflects saturated or near-saturated conditions. SP data shown in Fig. 9 were applied to Eq. (8), using $H_o/H_c = 1$ and parameters reported in Section 4.2 ($\lambda = 2.7$, $S_r = 0.15$, and $H_o = 0.098$ m) to calculate a flux in the lower 30 cm of the column, based on the SP measurements ($q_{sp}$). Zeta potential was then determined by calibrating $q_{sp}$ to $q_{meas}$ with a sum of residuals of zero over the test period, resulting in a value

of $\zeta = -0.2807525$. Water quality of the column effluent was measured, and during the time period shown in Fig. 9, pH ranged from 6.19 to 6.62, and $\sigma_w$ was equal to 10 μS (1E-5 S m⁻¹). The value of $\rho_s$ (416 kg/m³) was determined at the end of the column test by weighing the column and measuring the snow volume contained therein.

### 4.4 SNOTEL data

Fig. 10 shows the time series of air temperature, SWE, and snowpack depth measured at the SNOTEL station located 30 m

from the snow study plot. SWE and snow depth data were adjusted from the SNOTEL values to correct for a slightly deeper





snowpack at the snow study plot than the SNOTEL site. SWE and snow depth data are not available after June 13, when the SNOTEL station melted out. SNOTEL data were obtained from (https://wcc.sc.egov.usda.gov/nwcc/site?sitenum=602).

**4.5 In situ measurements**

In situ snowmelt measurements were collected from two SP monitoring intervals, an upper and a lower interval, as shown on

Fig. 6 and Fig. 7. The upper interval consisted of two SP electrodes, placed at 35 cm (E2) and 65 cm (E1) above the ground surface, without a $H_c$ tensiometer. The lower SP interval included two electrodes, placed at 0 cm (E4) and 30 cm (E3) above the ground surface, and a $H_c$ tensiometer spaced equally between the two electrodes. Fig. 11 shows the time series of SP data at the upper (E1E2) and lower (E3E4) SP intervals, and SNOTEL temperature data, collected over a 13-day period from May 28 through June 9, 2017. $H_c$ data collected from the tensiometer was unreliable prior to June 10, apparently due to poor contact

of the porous ceramic cup with the snowpack.

Fig. 11 shows that air temperature is a driver for diurnal changes in meltwater percolation. The upper SP interval (E1E2) (black line, Fig. 11) shows earlier response to downward movement of each diurnal wetting front, following the afternoon surface melting period. From June 1 through June 10, the velocity of propagation of the wetting front from the upper to the

lower interval ranged from 3.9 to 12.6 m d$^{-1}$, without an apparent correlation to the intensity of melting. Overnight temperatures were below freezing on May 28, 29, and 30, which would be expected to result in surface refreezing of the snowpack. The mV-level SP signals were negative in value at both SP levels during the morning of May 28, and at the upper SP level on May 29 and 30, which may reflect upward capillary-driven flow in response to surface refreezing (Iwata et al., 2010).

Fig. 12 shows SP and $H_c$ measurements for only the lower SP interval from June 10 through June 15, 2017. SP electrodes at the lower SP interval melted out on June 16, and the upper SP interval is not shown on Fig. 12 because it melted out on June 12. $H_c$ data measured using the tensiometer during this period were reliable, presumably as increasing meltwater production made improved contact with the porous ceramic cup of the tensiometer. A sharp wetting front was observed in SP and $H_c$ data during the afternoon of each day followed by gradually declining SP and increasing $H_c$ after passage of the wetting front. Peak

SP signals ($\psi_m$) ranged from 45 mV on June 13, to 121 mV on June 14. Overnight, SP decreased to a range of 20 - 30 mV. On days of maximum melting (e.g. June 14), $H_c$ approaches $H_o$ (~7 cm) at the peak of the wetting front, indicating saturated or near saturated conditions. Overnight $H_c$ values decreased to approximately 12 cm.

The calculated $q_{sp}$ values shown on Fig. 12 were determined for each 10-minute data acquisition time step, using Eq. (8) and

parameters reported in Sections 4.2 and 4.3. Parameter values applied included: $\lambda = 2.7$, $Sr = 0.15$, $\varepsilon = 7.8E\text{-}9$ F m$^{-1}$, $\zeta = -0.2807525$ V, and $\sigma_w = 1E\text{-}5$ S m$^{-1}$. Using Eq. (9) $k = 6.7E\text{-}9$ m$^2$, based on measured values of $d = 1.5$ mm and $\rho_s = 416$ kg/m$^3$. The value of $H_o$ (0.06977 m) was taken as the minimum value observed from tensiometer readings during in-situ measurements





in the time period of Figure 12. Peak values of $q_{sp}$ presumably occur as the snowmelt wetting front percolates through the snowpack, with values ranging from 5.6 cm/d on June 13 to 105 cm/d on June 14.

Fig. 12 also shows a comparison of $q_{sp}$ to changes in SWE at the SNOTEL station. The measurement of $q_{sp}$ at each time point

represents water draining from the bottom of the snowpack, as the lower interval of SP electrodes were placed in the 0.3 m interval above the ground surface. Therefore, if it is accurate, $q_{sp}$ should reflect changes in SWE measured at the SNOTEL station. SNOTEL SWE data are collected hourly, but the SNOTEL station had earlier sun hit by approximately one hour, so an hourly comparison of SWE and $q_{sp}$ is not useful. Therefore, Fig. 12 includes data for the 24-hour change in SNOTEL SWE ($\Delta SWE_{24}$) and the integral of SP-measured fluxes over a 24-hour period ($\int q_{sp-24}$). Both values were taken from midnight to

midnight each day. The SNOTEL station melted out on June 14, so $\Delta SWE_{24}$ was calculated for the four-day period from June 10 through June 13, 2017. Individual $\Delta SWE_{24}$ values ranged from 2.5 to 5.3 cm/d, as compared to individual values of $\int q_{sp-24}$ which ranged from 1.3 to 6.7 cm/d, indicating that errors in calculation of 24-hour flux ranged from +26% to -47%. Over the four-day period, the total change in SNOTEL SWE was 16.8 cm, as compared to the total SP-calculated meltwater flux of 18.1 cm. This represents an average error of +8% in the calculated values of $q_{sp}$ over the four-day period. Measurements of pan

evaporation during a previous snowmelt season (Hutchison, 1966), at a location approx. 30 km from the site, indicated an average of 0.1 cm/d evaporative loss, which is negligible relative to the melting rates observed herein.

Fig. 13 summarizes unsaturated flow parameters determined from the in situ meltwater percolation measurements, including $q_{sp}$, $S_w$, and $H_c$. These results could be used for calibration of an unsaturated flow model of the snowmelt percolation process,

although that was not undertaken herein.

**5 Discussion and conclusions**

An uncertainty pertaining to the results herein is that the in situ measurements are point measurements within a spatially variable flow field. While the values of percolation flux that were calculated scaled similarly to changes in SWE at the adjacent SNOTEL station, it is likely that in some circumstances substantial variability would be encountered point-to-point.

Measurement of SP in a 2-D gridded vertical profile could potentially shed light on the spatial variability of unsaturated flow processes in a heterogeneous snow pack.

An important limitation of the method pertains to the three-step process that requires a snow core sample test and a column test that are essentially contemporaneous with the in situ measurements. The constitutive properties such as $\lambda$ (Brooks and

Corey, 1964) and others measured in the core and column tests may vary spatially and temporally in melting snowpacks. For this reason, these tests may need to be repeated using multiple samples from different locations or times. It is important to note that zeta potential ($\zeta$) is a sensitive function of pH and conductivity of the meltwater (Revil et al., 1999). Herein, the





value of ζ was determined empirically from the column test, and then assumed to be constant over the 19-day in situ measurement period. However, changes in meltwater chemistry may have occurred over that time that could cause the value of ζ to drift. This problem may be less pronounced in glacier snowpacks that are thicker and have a more extended melt cycle than thin alpine seasonal snowpacks. Further evaluation of the sensitivity of ζ in association with in situ measurements may

be important in future research.

The research reported herein showed that mV-level SP signals and capillary pressure can be measured in situ, in a vertical profile configuration in melting snowpacks to determine downward percolation flux. The data showed a strong qualitative relationship to diurnal snow melt dynamics, and the calculated fluxes were quantitatively comparable to actual fluxes, as

represented by changes in SWE at the adjacent SNOTEL station. Thompson et al. (2016) evaluated sources of error in the various parameters involved in Eq. (1), and an error analysis was not undertaken herein. More rigorous testing is needed to evaluate the error associated with using SP signals combined with capillary pressure or saturation to calculate meltwater percolation flux. Nonetheless, errors ranging from +26% to -47% over a 24-hour measurement cycle, with an average error in calculated flux of 8% over a four-day period, is an encouraging result for this exploratory study.

The methods developed herein for in situ measurement of meltwater percolation flux involve data-logging SP electrodes, in conjunction with either saturation or capillary pressure sensors. Capillary pressure sensors were used herein for simplicity. However, challenges were encountered with maintaining good contact between the porous ceramic cup of the tensiometers and the snowpack, in both the column test and the in situ measurements. Given this challenge, future in situ measurements

may be best performed using SP in conjunction with TDR to measure saturation, and using Eq. (7) in place of Eq. (8).

The use of SP, in conjunction with either capillary pressure or saturation, to measure in situ flux of percolating meltwater in snowpacks is promising. The measurements could be used to support mathematical modeling of unsaturated flow processes, or to support measurements of snowmelt-groundwater or snowmelt-runoff interactions. The measurements also have potential

to supplement glacier mass balance studies by quantitative measurement of percolation flux within seasonal snowpack or firn.

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



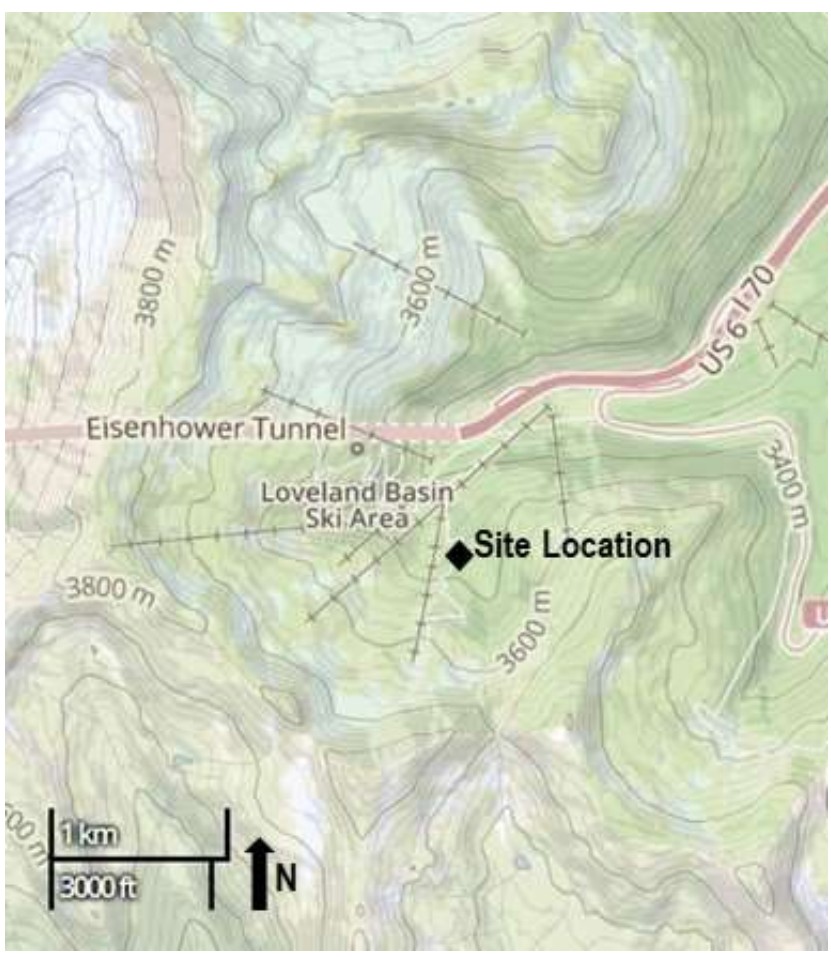

Figure 1: Location map of the snow study plot at Loveland Basin ski area. Map source: https://ngmdb.usgs.gov/topoview/

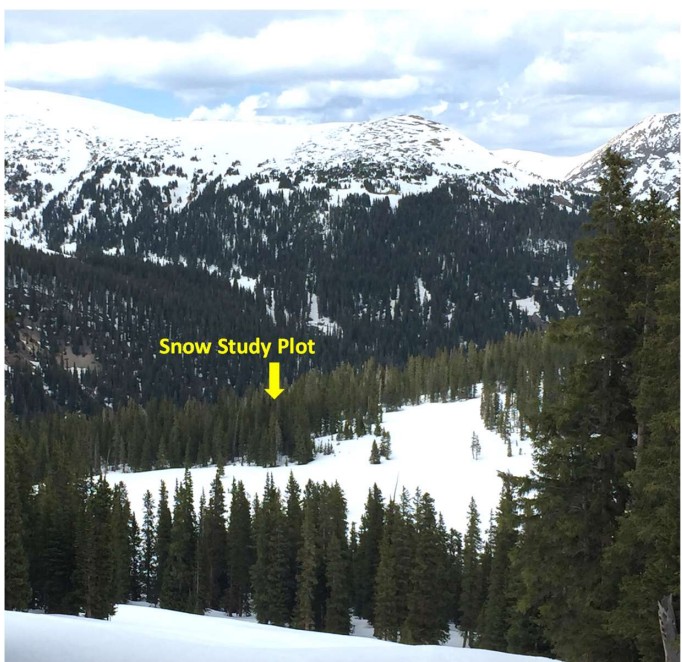

**Figure 2: Photo of the snow study plot location, looking north, taken on May 30, 2017.**

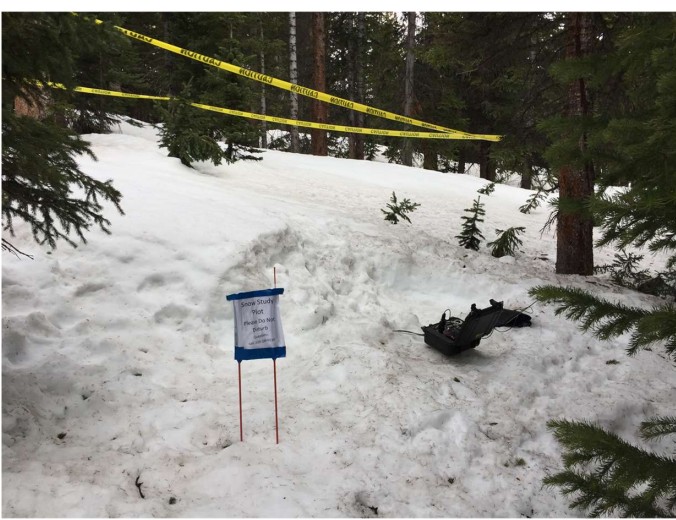

5    **Figure 3: Photo of the snow study plot, after installation of in situ measurement sensors and backfilling, on May 28, 2017.**





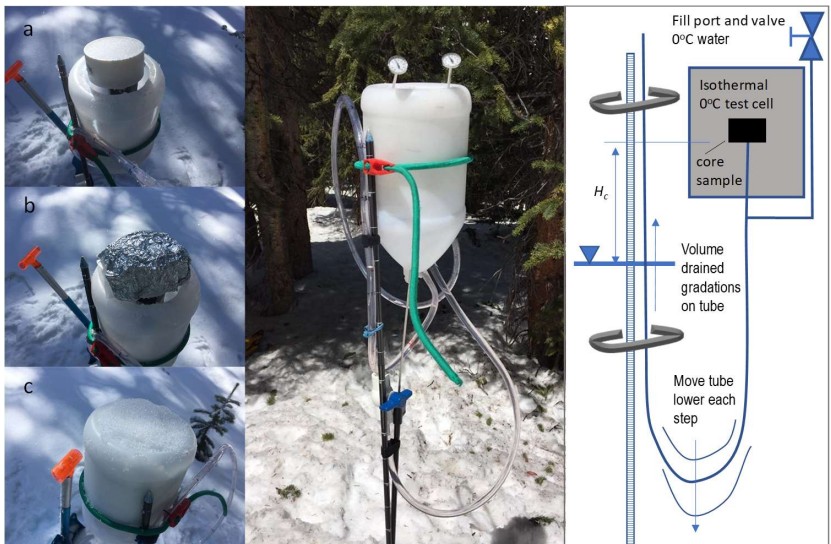

**Figure 4: Isothermal (0ºC) snow core sample holder used for measurement of drainage characteristics.  Series of three photos shows (a) the snow core sample which is then (b) covered in aluminium foil and (c) packed in snow within the outer chamber.  The plumbing and operating principle of the test cell is also shown, at right.**

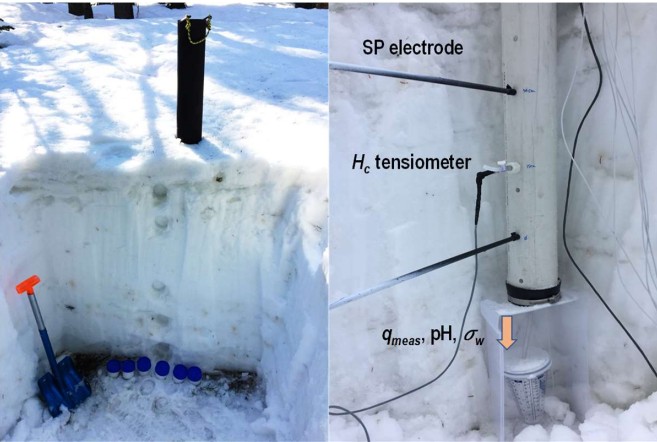

**Figure 5: Photo of snowmelt column experiment, showing (a) inserted but unexcavated snow column, and (b) excavated column with snowmelt testing ongoing.  Upper third of core tool is painted black to promote radiant warming by the sun.**





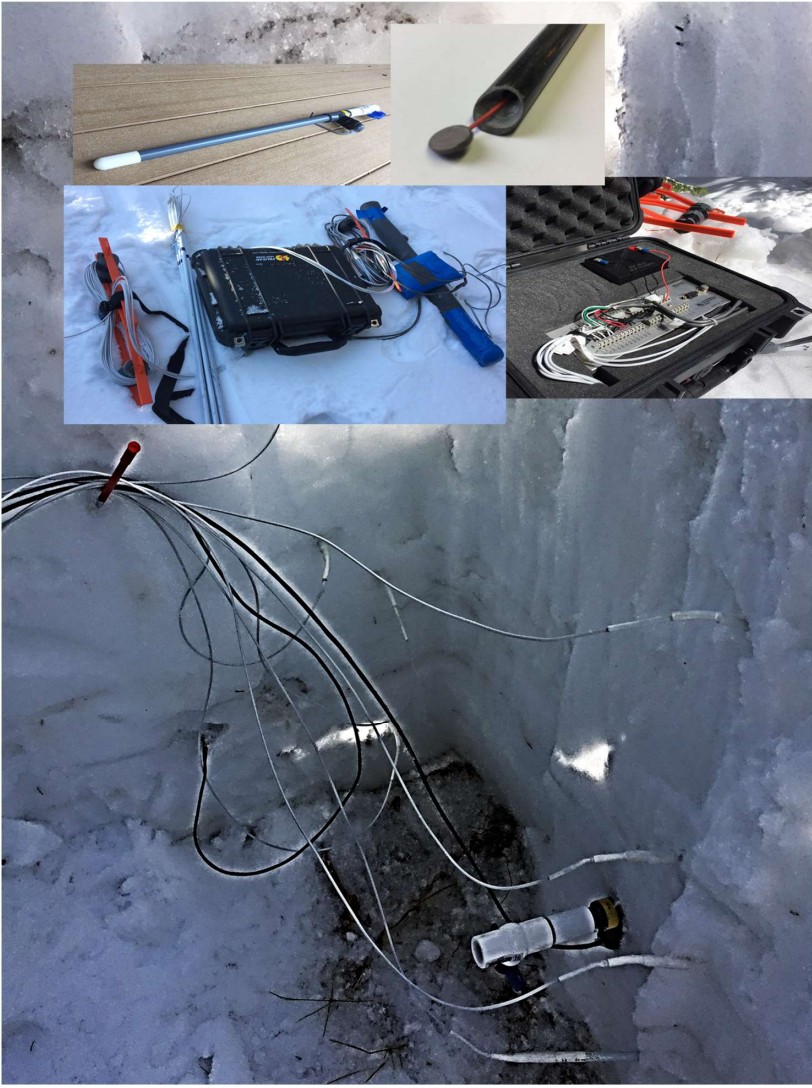

**Figure 6: Photo of SP electrodes and $H_c$ tensiometer installed in sidewall of snow pit, prior to backfill. Sensors on right hand side are shown in Figure 7 and were used for data analysis herein. Sensors on left hand side failed, presumably due to poor contact with the snowpack. Inset photos show SP and $H_c$ measurement equipment including (clockwise from upper left): 50 cm-long $H_c$ tensiometer, SP electrode tip in partially assembled state, CR-10 data logger in weatherproof Pelican Case, and assemblage of SP and $H_c$ sensors and instrumentation.**





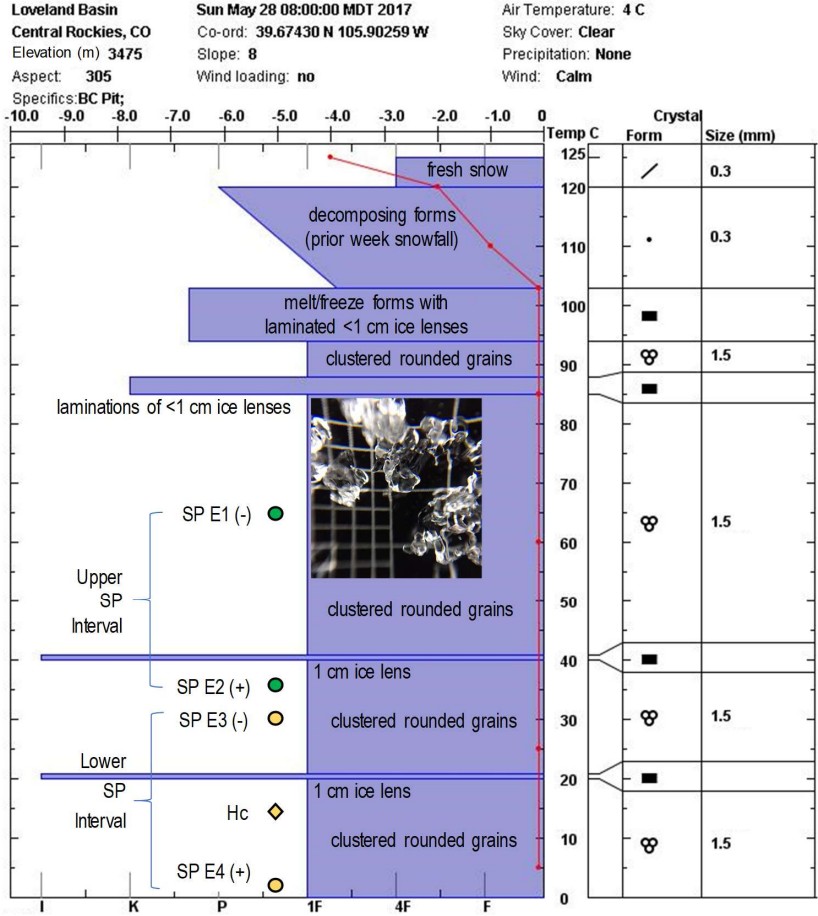

**Figure 7: Log of snow study pit on May 28, 2017 at 08:00. Inset photo shows snow crystal forms encountered from 0 cm to 94 cm, placed on grid marked at 1x1mm and 3x3mm. Temperature profile shows isothermal 0ºC snowpack below 104 cm, and frozen snowpack above 104 cm, at the time the pit was logged. Depth intervals are indicated for installation of SP electrodes and $H_c$ tensiometer used for data reported herein. Log prepared using SnowPilot software.**



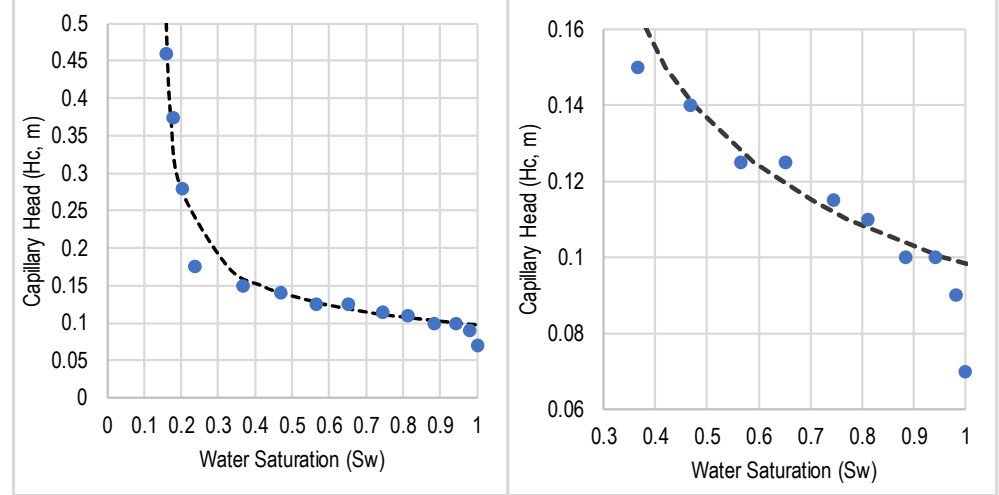

**Figure 8: Capillary pressure head ($H_c$) vs. saturation ($S_w$) for the snow core sample drainage test. The dashed line represents Eq. (6) fitted to the data with Brookes and Corey (1964) parameters, $\lambda$=2.7, $H_o$ = 0.098, and $S_r$ = 0.15. Graph at right is zoomed to lower values of $H_c$.**

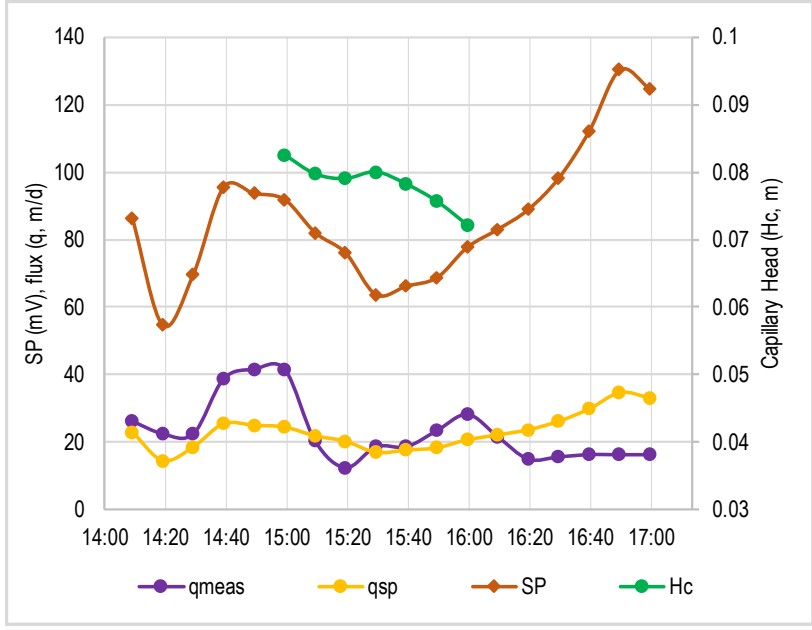



**Figure 9: Column snowmelt test results showing measured capillary pressure ($H_c$), measured SP, the flux that was eluted from the column ($q_{meas}$), and the SP-calculated values of flux ($q_{SP}$), which were calibrated using a value of zeta potential = -0.2807525, with a sum of residuals of zero.**

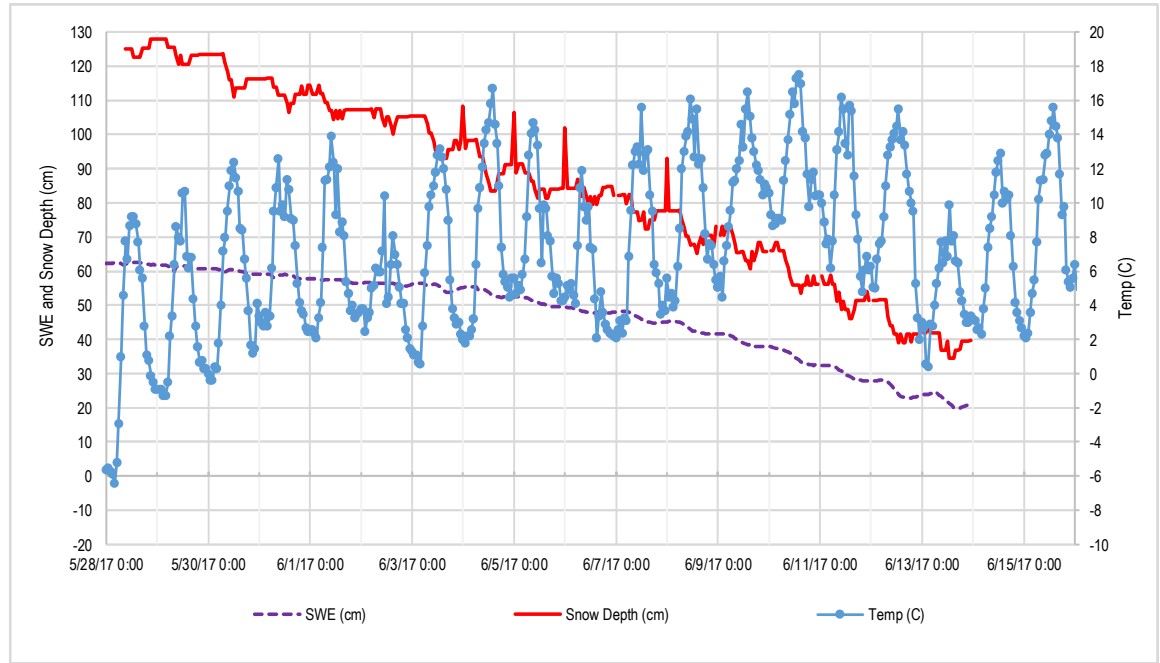

**Figure 10: SWE, snow depth, and temperature data from the SNOTEL station, located 30 m from the snow study plot. SWE and snow depth were adjusted from the SNOTEL values to correct for a slightly deeper snowpack at the snow study plot than the SNOTEL station. SWE and snow depth data are not available after June 13, the date that the adjacent SNOTEL station melted out.**





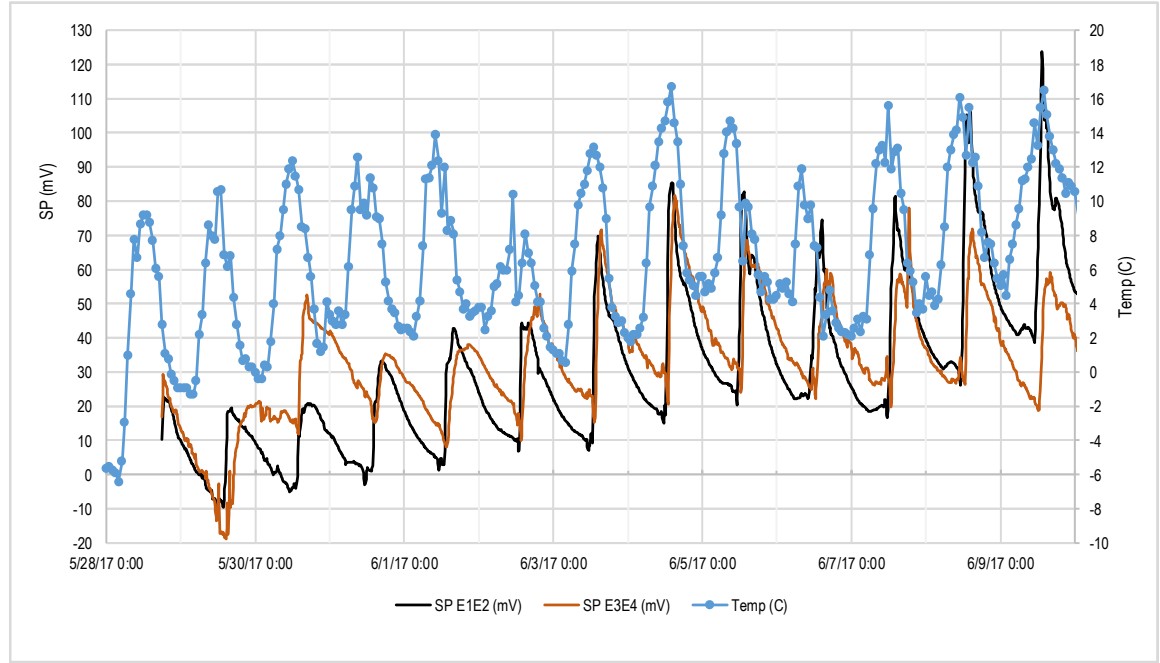

**Figure 11: In situ snowmelt SP measurement results from May 28 through June 9, 2017, shown with SNOTEL temperature data. Temperature appears to drive the diurnal melt cycle and the upper SP interval (E1E2) (black line) shows earlier response to each diurnal wetting front.**

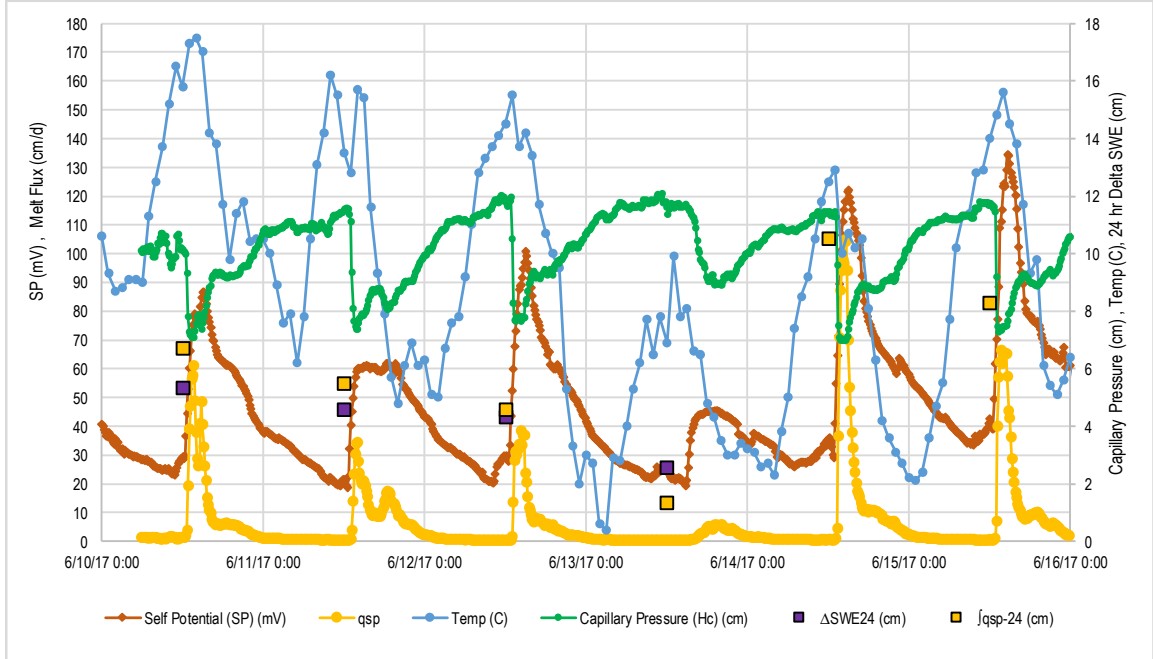

**Figure 12: In situ snowmelt measurements from June 10 through June 15, 2017. The value of $q_{sp}$ was determined using the SP and $H_c$ data as well as parameters measured in the snow core sample test and the snowmelt column test. The 24-hour change in SWE from the SNOTEL data was calculated through June 13, before the SNOTEL station melted out.**



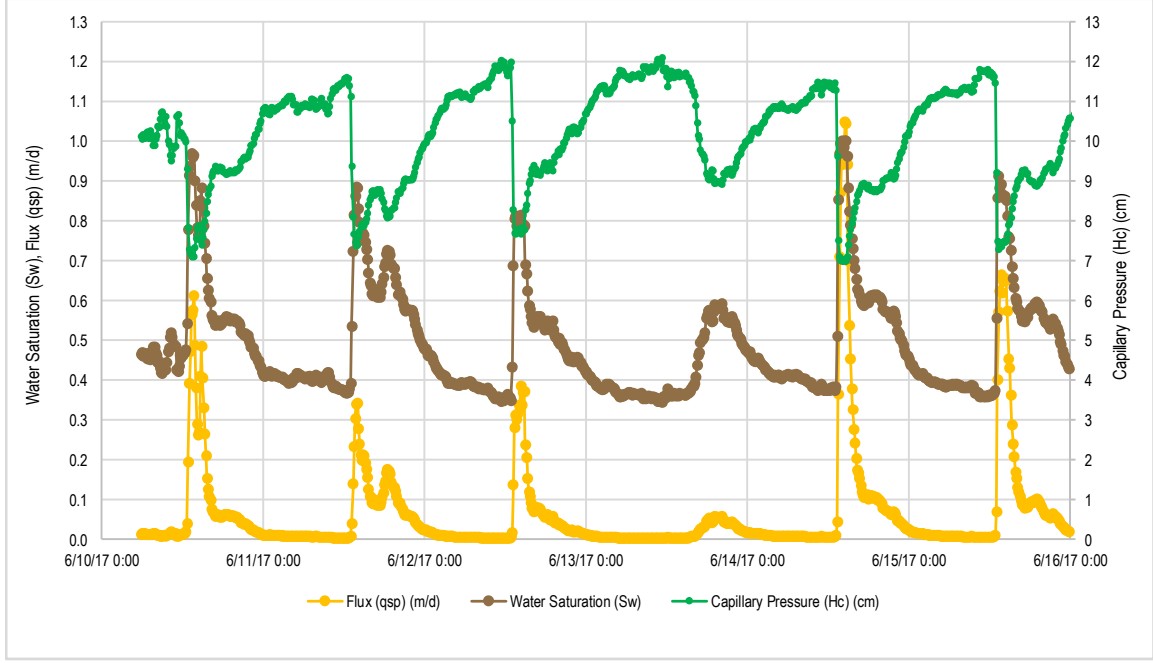

**Figure 13: Summary of unsaturated flow parameters, consisting of flux ($q_{sp}$), water saturation ($S_w$) and capillary pressure ($H_c$) for the period from June 10 through June 15, 2017. $S_w$ was calculated using measured $H_c$ and Eq. (6) and Eq. (3).**