# Peer review of "In situ measurement of meltwater percolation flux in seasonal alpine snowpack using self potential and capillary pressure sensors"

_The Cryosphere, 2017_

## Referee Comment (RC1) · Anonymous Referee #1 · 17 Nov 2017

The manuscript "In situ measurement of meltwater percolation flux in seasonal alpine snowpack using self potential and capillary pressure sensors" by Clayton presents a case study with new instrumentation to determine melt and liquid water within seasonal snow. The author describes a self-constructed, transportable instrumentation applicable for conventional fieldwork without transportation support. Measurements with the described system are destructive, which in consequence have a number of limitations and are prone to certain errors. The presented manuscript is well written, it presents a novel approach and might contribute to validation of simulation outputs. However, apart from the assembling and combination of the applied measurements, the presented data are limited to a time period from a single day to maximum two weeks. It

remains a bit unclear, how this work contributes/ improves our knowledge, while being compared with already existing methods such as capacity plate sensors (Denothmeter) or TDR sensors, the usage of lysimeters for capturing outflow or the cited work on self-potential signatures.

Before publication, I recommend to clearly state the purpose of this work and describe the benefit for the scientific community in comparison to previously published approaches/ data sets. Right now, it is just a feasibility study, which produced deviations to conventionally measured data of almost up to 50%. The current status of this manuscript does not rectify publication within The Cryosphere. In addition, the presented data must be compared with long-term observations of liquid water content behavior, outflow and diurnal changes thereof in seasonal snow presented by Heilig et al. (2015 – doi: 10.1002/2015JF003593). Such measurements were conducted over several months for four consecutive years at three different locations and in a non-destructive manner. Hence, stratigraphy remained undisturbed by measurements and instrumentations. Since you did not present a single data set on snow density/ porosity, it is very difficult to compare the presented results on diurnal changes and saturation with previously published data. Furthermore, you certainly should include work published by F. Avenzi and colleagues about model assessments and measurements of liquid water in seasonal snow. The reference of Samimi and Marshall is just the most recent one for TDR but for introduction of such methodology you rather have to cite Sihvola and Tiuri (1986) and Schneebeli et al. (1998). Avenzi clearly demonstrated that such probes (which you are using in a similar way) can be affected by melt out, heat conduction via cables and air voids surrounding the sensors. For a long-term monitoring this is a strong limitation. The spatial support of your measurements are very limited to just the placements of the sensors. A comparison with snow pillows and ultrasonic transducers being located in a distance of 30 m is a least questionable (i.e. think about heterogeneous percolation). Please include into discussion how much the zeta potential can vary, any literature data, measurements you conducted? This will actually allow for determination of error ranges.

Some minor points that need to be revised:

P2 L29 give units for the area (m2)

Don't use a point for the abbreviation of meter P5 L7

Please state how far apart from each other all three measurements were conducted (P5 L12ff)

P6 L13 please present data on density determinations

P8 L12 well, it is actually the full energy balance at the location that drives snow melt. Air temp. might be a result from the radiation budget or might be laterally transported by sensible heat, released as latent heat etc. Please be precise.

P9 L9ff it might be more appropriate to use 6am for determination of diurnal changes since at that time usually daily minimum in temperature is reached. (see Heilig et al., 2015)

Sihvola, A.; Tiuri, M. Snow fork for field determination of the density and wetness profiles of a snow pack. IEEE Trans. Geosci. Remote Sens. 1986, 5, 717–721. Schneebeli, M.; Coléou, C.; Touvier, F.; Lesaffre, B. Measurement of density and wetness in snow using time-domain reflectometry. Ann. Glaciol. 1998, 26, 69–72.
* * *

---

## Referee Comment (RC2) · Anonymous Referee #2 · 28 Nov 2017

GENERAL COMMENTS

This manuscript with the title 'In situ measurements of meltwater percolation flux in seasonal alpine snowpack using self potential and capillary pressure sensors' by Clayton deals about a new experimental study of three measurement steps at a high-alpine test site in central Colorado near a permanent SNOTEL snow monitoring station. With this feasibility study, the author aims to improve the understanding of liquid water within a snowpack which might be useful for a better understanding of snow melting processes. The topic of this manuscript is interesting, however, the paper in its current form is not recommended for publication in The Cryosphere – I suggest major revision.

[Figure]

I agree in all main points with Referee 1. This encompasses that it has to be clearly stated that it is a feasibility study and that the entire measurement time period is quite short and should actually be repeated (this is the case for all three measurement steps) and the in situ measurement step should be longer to derive more reliable results. The literature review should be enlarged / completed regarding classical (dilution, calorimetry, permittivity) and modern techniques (e.g. radar, GPS), which can continuously track diurnal and seasonal variations in liquid water, and whereof some are even non-destructive. It should be emphasized to a greater extent that this is a destructive method which is by nature error prone, especially as flow paths and snow pores as well as the temperature gradients within the snowpack around the installed sensors are very likely different to the conditions in the undisturbed snowpack – please include also relevant literature references on this issue.

SPECIFIC COMMENTS

SWE: The unit of SWE is millimetres – please correct this in the text and the figures. This will also help to distinguish between snow depth and SWE, which is not always clear in the manuscript. This is also the case when you introduce q_SP with cm/d – I think it should be mm/d. LWC: Is it possible to additionally derive the volumetric liquid water content (LWC) in snow with your method (as you can do so with tensiometers for soil moisture measurements if you know the soil texture and by applying a water retention curve)? If so, this would increase the value and comparability of this study.

Rearrangement of sections: The introduction section should give a broader overview on the state of the art and it I would recommend to already mention here your objectives (currently section 3). The passages where you start to introduce equations should be shifted to the methods section (e.g. as a sub-section). Some statements in the results section already belong to the discussion part – please separate results and discussion clearly or rename the section to 'results and discussion' and avoid repetitions (e.g. p.8, l. 31). The discussion has to be improved markedly and I would suggest to separate it from the conclusions.

References: Besides the additional references Referee 1 mentioned, also work from Schmid et al. (doi: 10.1002/2015GL063732), Koch et al. (doi: 10.3390/s141120975) and Bradford et al. (10.1029/2008WR007341) should be mentioned in the introduction section. Regarding capacity measurements like Snow Fork and Denoth meter measurements, Techel & Pielmeier (doi: 10.5194/tc-5-405-2011) presented a very comprehensive study. Regarding discussions on the diurnal patterns on liquid water in snow, also recent work from Webb et. al (e.g. doi: 10.1111/1752-1688.12522) should be mentioned.

Figures: The quality of almost all figures has to be improved.

In the abstract you abbreviate 'self-potential' with SP, however, later in the manuscript (p.2. l.16) you abbreviate 'steaming potential' with SP. Please use this abbreviation in a uniform way – this might also have an influence on the wording in the title. Moreover, if you want to keep 'self-potential', please correct the title to self-potential with '-'.

Can you state on the issue that during snow melt, the sensors might be melted out and are lowered in their installed position height? Actually I can't imagine that you recorded useful data with sensors SP E1 or SP E2 at the end of your measurement period.

P2, l.22: Please correct 'dialectric' to 'dielectric' and it is not clear what you actually mean. Is it the relative permittivity, the real part of the complex permittivity of soil or the permittivity of free space (e0) - but then the value would not be correct? Please specify.

p.3, l.6: You write 'we' – do you have co-authors? If so, please mention them or change the text. p.3, l.10 'q' should be written cursive – please check if all variables are written in a uniform way (also in the figure legends). This is for example also an issue for the variable 'Se' in various equations, etc.

p.5, l.7: You write that your test site is covered with 'mixed and forested shade' and that it is installed in a slope with an angle of 8° – is this comparable to the conditions

at the SNOTEL site? Please give more information on the SNOTEL site. Actually the conditions at your test site seem to be different, as the SNOTEL station melts out earlier.

p.8, l.11: (5) Include information on the radiation or even the full energy balance as temperature describes not the entire melting processes.

p.9, l.2: 105 cm/d (1050 mm/d) seems to be very high – is this value realistic?

p.9, l. 8-12: Is it really possible to compare the bulk snow melt values, which you can measure with the snow pillow with your specific point measurements at the sensors installed at a certain snow depth. Please state on this in more detail.

p.9, l. 19/20: This is an interesting point, but should be given as an outlook – instead of in the results section.

p.10, l.13: Actually, an error range of +26% to -47% seems to be not too encouraging. Please state on this.

p.12, Figure 1: It is nice to include an overview map, However, this one is too imprecise. An overview should also include the SNOTEL station.

p.13, Figure 2: Actually, in this picture you cannot really see the test site or to which extent it is covered by trees. Maybe you can include a picture in Figure 1 as a small picture and another one showing the SNOTEL site.

p.13, Figure 3: Not sure if this picture is really necessary either. Maybe it can also be included in Figure 1.

p.16, Figure 7: The first 5 cm are described as fresh snow, however, a grain size of 0.3 mm is very unusual for fresh snow, this might also be wrong for the decomposed forms. Moreover, the symbols of those two layers are not correct for the form you describe in the blue part.

Figures 9 – 13: In general, the legends and axis texts are very inhomogeneous. Some

legend items are written as abbreviation, some include the full wording, some include units, some do not. Units should not be included in the legend - they are part of the axis text. Please be uniform and remember that a figure should be self-explanatory, which is presently not always the case.

Figures 11-13: What do the straight brown or green lines, respectively, show? It looks like a display error, please correct.

Figure 11: Please insert that you show the mean of SP E1E2 or SP E3E4, respectively. Why do you actually show the mean of two sensors – it would be interesting to see E1, E2, E3 and E4 separately, as they are all installed at different heights.

Figure 12: The values of yellow boxes (sum of q_SP-24) of the last two days are very high – is this realistic?

Figure 13: This plot contains the same information than Figure 12 – it seems unnecessary and redundant. Moreover, the units should always stay the same (this is e.g. not the case for the melt flux). Moreover, to stay uniform, you should also insert the unit for water saturation in the y-axis (instead of its abbreviation).

---

## Editor Comment (EC1) · F. Dominé (Editor) · 4 Dec 2017

Dear Dr. Clayton

After carefully considering the evaluation of both reviewers, I am sorry to come to the conclusion that your manuscript cannot be accepted for publication in The Cryosphere with the amount of data provided. Both reviewers agree that the data you provide is not sufficient as it has been obtained over a very short period during just one melt season, while previous studies on similar topics studied much longer periods, and the studies often extended over several years. Your work may have been considered as a feasibility study, but reviewer 2 casts serious doubt about the validity of your methods

and points to references which discussed difficulties with your experimental choices. Furthermore, your discussion of previous work, and of the added value of your system, appears insufficient.

In summary, a much larger data set, a better justification of your technological choices, and an extended discussion of the interest of your system in light of previous studies would have been required. Performing the necessary improvements within the time allowed for revisions does not appear realistic.

I wish you the best of luck in your future studies, and we hope to have the opportunity to consider suitable manuscripts from you in the future.

Best regards

Florent Domine

Co-Editor-in-Chief

―――――――――――――――――――――

---

## Author Comment (AC1) · 22 Dec 2017

As the author, I would like to thank the two anonymous reviewers and the editor for many comments that will be useful in future work. Responses to individual comments are not provided here. The manuscript represented a rapid communication of exploratory research, and I acknowledge the comments that additional data and better constraints on measurement errors are needed. I understand the decision to reject the paper on this basis.

A significant number of reviewer's comments pertained to a lack of addressing prior work in measurement of liquid water content in snow. It is worth clarifying that liquid

water content was not the subject of the manuscript. The manuscript described vertical meltwater percolation flux (i.e. Darcy velocity) in the snowpack.